# The organizational production of earnings inequalities, Germany 1995–2010

**Donald Tomaskovic-Devey**[1]*, **Silvia Maja Melzer**[2]

**1** Department of Sociology, University of Massachusetts, Amherst, MA, United States of America,
**2** Department of Political & Social Sciences, University Pompeu Fabra, Barcelona, Spain

* tomaskovic-devey@soc.umass.edu

**Data Availability Statement:** This study used third-party data which are highly confidential under German law. Data are available from the German Federal Employment Agency (Bundesagentur für Arbeit) and provided by the Institute of

## Abstract

Germany has experienced sharply rising earnings inequalities, both between and within workplaces. Working from prior literature on rising employment dualization and the fissuring of workplaces into high and low wage employers, we explore a set of organizational explanations for rising between and within workplace inequality focusing on the role of employment dualization, skill segregation/complexity, and firm fissuring. We describe and model these hypothesized processes with administrative data on a large random sample panel of German workplaces. We find that rising inequalities are associated with polarization in industrial wage rates and the birth of new low wage workplaces, as well as increased establishment skill specialization and the growth of part-time jobs in workplace divisions of labor. We conclude with recommendations for future research that directly examines more proximate mechanisms and their relative importance in different institutional contexts.

## Introduction

Economists, organizational scholars, and sociologists [1–3] have advanced strong theoretical and empirical arguments that social scientists should increase their focus on the role of workplaces in generating earnings inequalities. Recent research has discovered that in multiple countries rising earnings inequalities have been associated with between workplace polarization in wage rates [4–6], underlining the importance of developing robust analyses of workplace inequality processes. The main empirical contribution of this paper is to develop and explore a set of available organizational explanations for rising between and within workplace inequality, focusing on the role of employment dualization [4], workplace and industry skill segregation [5], and firm fissuring [7]. A second contribution is to develop an original research design to examine workplace inequality dynamics.

We take advantage of fifteen years of German administrative data (1995–2010) on a large random sample of workplaces and all of their employees. Germany is the largest economy in Europe and the fourth largest in the world; it is also an important case in that earnings inequalities have risen rapidly both between and within workplaces [4, 6]. Our estimates [6] are that by 2010 total inequality rose a remarkable 31% over its level in 1995. Within workplace earnings dispersion increased 22.7%, while between workplace earnings dispersion surged by 56.8%.

Employment Research to researchers who meet the criteria for access to confidential data. Access to the data is available under licence through the German Institute for Employment Research (IAB) [https://www.iab.de/en/ueberblick.aspx]. This data and similar data sets are available for scientific analysis upon filing an application [https://fdz.iab.de/en/FDZ_Data_Access/FDZ_Scientific_Use_Files.aspx#apply]. The authors did not have any special access to this data.

**Funding:** DTD was supported by Alexander von Humboldt (grant AR8227) and U.S. National Science Foundation (grants SES-1528294; SES-1852756). The funders had no role in study design, data collection and analysis, decision to publish, or preparation of the manuscript.

**Competing interests:** The authors have declared that no competing interests exist.

The most prominent explanation for rising inequality in Germany focuses on institutional shifts–particularly the birth of new non-union firms and the rise of part-time labor–both of which potentially undermine the bargaining power of labor more generally [4]. We directly model the impact of both establishment births and part-time labor on rising within and between workplace inequality. In addition, we explore a novel explanation of institutional change that focuses on corporate strategy and firm fissuring, in which firms with market power outsource production roles, particularly those of lower skilled labor, leading to rising between firm skill segregation [5, 7]. We see these two explanations as complementary: declining institutional protections make it easier to outsource, while outsourcing can also lead to declining collective worker power.

There has been very little research that makes the direct connections between organizational processes and rising between or within workplace earnings inequalities. In this paper we review what has been speculated on and what is known in this regard and derive a collection of hypotheses that are consistent with the variety of underlying mechanisms identified in the literature. Some of these hypotheses are at the societal level and tend to receive fairly strong support. Others are at the workplace level, and while our evidence is mixed it generally confirms that changes in skill composition and rising dualization in job structures are important contributors to both between and within workplace inequality dynamics. While these exercises are far from definitive, we see them as strongly suggestive of which mechanisms deserve to be more fully explored in the future.

We find that between workplace inequality dynamics are associated with industry level wage polarization, the birth of new low wage workplaces, and reductions in occupational skill complexity. We also confirm the dualization hypothesis that the rise of part-time work increases the bargaining power of full-time labor within workplaces, while decreasing average wage rates, even among full-time workers, between workplaces. Disconfirming our expectations, within workplace inequality is higher in both high and low wage establishments and rises when earnings both fall and rise. Stable and average pay establishments are the most internally egalitarian.

## Previous research

Early research using linked employer-employee administrative data for multiple countries found that most earnings inequalities were within, rather than between, workplaces [8, 9]. Recent research, in contrast, has discovered that a great deal of rising inequality is a between firm phenomena [4–6, 10]. Our recent study using administrative data for fourteen countries finds that in twelve countries the share of total inequality that is between workplaces was rising, that in six the between workplace inequality component is now larger than the within establishment component, and that these trends were associated with declining institutional protections of low wage labor [6].

Past research has found that increased between workplace earnings inequalities produced about 60% of growing West German male wage inequality, two-thirds of which was associated with increased occupational skill segregation between workplaces [4]. As likely explanations for these trends the authors point to the birth of new non-union firms as well as labor market reforms which permitted German employers to create part-time and fixed-term employment contracts, and to outsource jobs to labor contracting firms. We directly examine establishment births and the rise of part-time jobs. We do not observe outsourcing or fixed-term contracts directly but discuss their likely influence.

Rising inequality has also been linked to changes in organizational structure. The disintegration of large, dominant, vertically integrated firms implies rising national inequalities as

high wage/low inequality firms shrink as employers relative to low wage/high inequality firms [11]. This process, it is argued, has been encouraged by the shareholder value movement, which has shifted large firm goals from employment, production, and market share to a focus on core competencies, lean production, and return on investment. This aspect of financialization encourages dominant firms to externalize as much of production and administration as possible, leaving the economic rents associated with their market and brand power for CEOs, shareholders, and fewer high wage employees, while simultaneously creating subordinate low-wage trading partners [7].

Consistent with this account, in the U.S. the largest firms accumulated larger shares of the value produced in the economy, even as their share of employment shrank [12]. From this theoretical vantage, between workplace earnings polarization is produced by some firms becoming more powerful in their market positions and so accumulating larger shares of national income, while simultaneously reconfiguring organizational boundaries to outsource routine production and support functions, moving from vertically integrated to market-based sourcing of intermediate products and services.

Examples of these externalization processes include manufacturing giants spinning off labor to cheaper, dependent supplier firms [13], branded companies subcontracting out most low skill labor while absorbing the profits associated with the brand [7], global commodity chains in which routine production are sourced from low wage economies by firms in high wage countries [14], and growing large firm market power [15, 16]. There is little evidence at this point linking these processes to rising inequality.

There is evidence, however, that outsourcing tasks to low wage industries leads to reduced wages for the outsourced tasks. In Germany, outsourcing jobs to service industries is estimated to reduce pay by 10 to 15 percent, even though the work is done by largely the same people at the same site [17]. In the U.S. estimated wage declines were 4–7% for janitors and 8–24% for guards outsourced from manufacturing to service industries [18]. There is also evidence that U.S. firms are simplifying their divisions of labor over time [19]. Also for the U.S., recent research suggests that growing occupational homogeneity among employers is particularly acute among low wage workers, and that increased workplace occupational homogeneity can explain a large portion of rising U.S. earnings inequality between 2002 and 2016 [20].

If these processes are operating in Germany, we would expect that much of the rise in German between establishment inequality will be a function of polarization in incomes associated with the industrial location of production, and of growing occupational skill segregation between firms. If high earnings/high skill firms externalize low skill and non-core jobs to firms in other industries, this should intensify industry pay polarization. When a manufacturing firm externalizes its human resources, cleaning, and food service functions, those jobs relocate to firms in other, often lower wage, industries (business services, cleaning services, and food services, respectively). Similarly, simplifications in occupational divisions of labor at the firm level via the outsourcing mechanism should intensify between workplace inequality, while reducing within workplace inequalities in high income firms.

Although Germany has a history of industrially coordinated labor markets, it has been moving since the mid-1990s toward a more decentralized and globally oriented industrial relations system [21, 22]. Solidaristic bargaining across industries has given way to dualized bargaining structures between industries: weak unions and low collective bargaining coverage characterize low wage, typically service sectors, while strong unions in export-oriented manufacturing preserve their workers' high wages. These shifts are associated with both gender and contract status (part-time, temporary worker) employment segregation within and between firms. These dualization processes have also been hypothesized to be associated with the rise of new low wage, non-union employers in Germany [4].

This discussion leads to a linked set of hypotheses that obtain at different levels of analysis, including economy-wide trends over the study period, between-workplace differences in levels (e.g. means and standard deviations of organizational characteristics); and within-workplace changes in those levels over time. We refer to differences in levels in terms of more or less, or greater or lesser. With respect to changes over time, we use the words rising and declining, or increasing and decreasing.

We begin with four linked sets of hypotheses:

## Industry hypothesis

*H1. Economy wide between industry wage polarization should increase.*

## New establishment hypotheses

*H2a. New workplaces will have lower mean wages and lower skill profiles than existing workplaces*

*H2b. New workplaces will have lower internal inequalities and lower skill complexity than existing workplaces*

## Occupational structure hypotheses

*H3a. Economy wide, total between workplace variance in skill levels will rise*

*H3b. Economy wide, total within workplace skill complexity will decline*

*H3c. Between establishments, a higher skill level will be associated with a higher mean wage. Within establishments, an increasing skill level will be associated with an increasing mean wage.*

*H3d. Declining workplace skill complexity will be associated with decreased within workplace inequality*

Dualization of the labor market can happen within, as well as between, workplaces. The rise of fixed term contract work and part-time work are prominent practices in Germany. The German Hartz reforms of the early 2000s, which loosened regulations on both part-time and fixed-term contract work, is predicted to increase the incidence of firms building their labor process around part-time labor [22]. Others have speculated that growing part-time labor will undermine the bargaining power of labor more generally [4], but do not specify if this is a within or between workplace prediction. Somewhat in contrast, other scholars have stressed the utility of these types of precarious work to protect the wages of the remaining full-time "core" workers [23, 24]. For between workplace inequality, we follow the reasoning in [4] that part-time work undermines the bargaining power of labor. For within workplace inequality, we follow the reasoning in [22–24] and hypothesize that the rise of part-time work will produce dualized workplace structures that strengthen full-time employees' earnings claims.

## Hours dualization hypotheses

*H4a. Economy wide between workplace variance in part-time work will rise*

*H4b. Firms with more part-time work will have lower full-time wage levels, net of occupational skill levels and complexity*

*H4c. Rising part-time work will increase within establishment full-time worker wages*

Finally, the literature on the outsourcing of low wage employment by high wage firms [7] leads to the expectation that both high and low wage firms will have declining internal inequalities as they specialize in high and low skill/pay production regimes. In addition, the observation that high wage firms also tend to be low inequality firms [11] leads us to the expectation of particularly low inequality in high wage firms.

## Wage dualization hypotheses

*H5a. Both rising and declining workplace mean earnings will be associated with declining internal earnings inequalities*

*H5b. High wage firms will tend to have the lowest levels of internal inequalities*

## Additional explanations

Some scholars of rising between workplace inequality have interpreted the rising between workplace trend as reflecting individual level skill segregation between firms [5]. Our hypotheses follow previous research on Germany [4] and the fissuring literature [7] in emphasizing occupational, rather than individual skill, segregation. One prominent theory of organizational inequality emphasizes that gender and citizenship can operate similarly to education, channeling the distribution of rewards in workplaces, although this approach also prioritizes the role of occupational divisions of labor [25, 26]. We include educational, age, tenure, gender, and citizenship composition in our models as potential additional drivers of organizational inequalities.

## Materials and methods

We analyze a random sample of administrative data drawn from the social security records of the German Federal Employment Agency (Bundesagentur für Arbeit) linking employees to their workplaces over time. Although the data were anonymized prior to analyses, they are highly confidential under German law and must be analyzed in a secure data facility with a license from the German Institute for Employment Research (IAB) [https://www.iab.de/en/ueberblick.aspx]. This data and similar data sets are available for scientific analysis upon filing an application [https://fdz.iab.de/en/FDZ_Data_Access/FDZ_Scientific_Use_Files.aspx#apply].

The basis for the data is the integrated notification procedure for health, pension, and unemployment insurance, which came into effect in 1973 and was extended to cover Eastern Germany in 1991. Employers are required to submit notifications at least once a year to the responsible social security agencies concerning all of their employees covered by social security and earning more than a minimum wage threshold, which was 450€ in 2015. These data are of very high quality as they are used to track income and employment for social security and tax purposes.

Excluded from the data are Beamte–a class of permanent civil servants–as they do not participate in the national social security system. In the early 1990s, Beamte were 6.7% of all employees, declining to 5.2% in 2011. In addition, freelancers and the self-employed, who are also not covered by national social security, are not present. These make up about 11% of the labor force in 2009, up from 9% in 2005. Like Beamte, they tend to earn more on average than employees in the social security system [27], although they are much more heterogeneous and include many low earners. Thus, our sample misses two types of relatively advantaged workers.

As a result, we probably underestimate the levels of earnings inequality, although the impact on trends is unknown as Beamte are declining and the self-employed are rising.

The data were sampled by the Institute of Employment Research (IAB) from the Integrated Employment Biographies Sample (IEBS) combining individual records of employment (BeH) and benefit recipient history (LeH) [see 28 for a description of the source data]. Our sample covers roughly 5% of the German employee population and 20,000 establishments with at least one year of existence between 1994 and 2010. Although national data are available starting in 1991, the East German administrative data only become reliable in 1993. The period of great growth in German earnings inequality happens within our observation period. As we use a lagged imputation strategy to deal with earnings top coding, our estimations start in 1995.

Workplaces are observed via an establishment ID and are for the most part unique stand-alone workplaces. When a firm owns two or more workplaces operating in the same industry in the same municipality, they are reported under the same ID. When a workplace is sold to another firm, it will get a new ID. We treat such workplaces as new establishments [see also 4]. These data include both private and public-sector establishments from all industries.

Our sample mirrors the dynamic population of German establishments. Sampling occurred in two steps. In the first, 20,000 establishments were selected proportional to their number of job-years between 1994 and 2010. Smaller and shorter-lived establishments are selected with decreasing probabilities. We limit the maximum of the sampling probability to 0.3, as otherwise large workplaces would be drawn nearly completely into the sample, violating our confidentiality agreement. Compared to more typical sampling strategies this method prioritizes jobs as employment spells, rather than people or organizations as discrete units. In a second step, all employees of the selected 20,000 establishments were drawn from the IBES. For large establishments the number of employees was limited to 1,000 randomly selected employees, again to conform to confidentiality restrictions.

All analyses use sampling weights to increase the weight of larger establishments and under sampled employees to correspond to their population frequency. Prior to aggregation to the workplace level, this weighted sample reproduces all individual and organizational population characteristics. After aggregation, we drop any workplace that averages less than 20 employees per year observed. This produces a sample that is representative of all German workplaces larger than 20 employees and all of their employees between 1994 and 2010. Compared to the entire economy, the only population level shift of note is that our 20+ employee sample has a 0.4 year longer average workplace tenure than the full economy average.

We then aggregate all indicators to the workplace level, focusing our analyses on organizational variation in the mean and standard deviation of logged daily earnings of full-time employees. Most workplace characteristics are calculated on full-time workers only. The one exception is the workplace distribution of part-time employees, which is calculated from all employees. Thus, while we mainly focus on earnings inequality levels and trends among *full-time employees*, we examine if the growth of part-time employment influences full-time earnings both between and within workplaces, per hypotheses H4b and H4c.

## Dependent variables

Both of our dependent variables are aggregate versions of the log of real daily earnings for full-time employees. Earnings includes all earnings from the employer in the year, including bonuses and overtime, divided by days worked. We are missing the exact hours worked and it is primarily for this reason that we limit our analyses to full-time employees. Between workplace inequality is measured as workplace mean logged daily earnings. Hypothesis 1 predicts that rising between workplace inequalities will be linked to rising between industry wage

polarization. Hypotheses 2a, 3c, 4b, and 5b also relate to between workplace inequalities, and are estimated regressing mean workplace earnings upon organizational level predictor variables. Within workplace inequality is observed as the standard deviation of daily earnings within workplaces and this is the measure responsive to hypotheses 2b, 3d, 4c, and 5a.

The key limitation of the earnings measure is that it is top-coded at the maximum income subject to German social security withholding. We improve on an existing top-code imputation strategy by using workplace as well as individual level information and prioritizing information from the two years previous to the top-coding of the wages to impute the income of higher earners (see S1 File for technical details and code).

## Predictor variables

Industry is measured with the three-digit standard industry scheme and is used to predict economy wide change in between and within workplace inequality trends. We only have a hypothesis (H1) with regard to the former, but include the latter for comparative purposes. Although there are some changes in industry coding in this period (see Table 1), they do not impact our estimates.

We observe the birth of new establishments as any workplace which appears after the first year of the time series. We hypothesize that new workplaces will tend to pay lower wages than existing workplaces (H2a) and also have lower skill complexity than existing workplaces (H2b). It is also possible that new establishments have lower wages simply because they are more likely to be economically marginal. To partially account for economic marginality, we also include an indicator of any establishment that exits the panel prior to the last year. If we are actually seeing a shift to low wage firms, the wages of new firms should also be lower than that of the firms that died. This is not a time varying indicator. Of course, these are imprecise measures since exits can be produced by other factors, such as being acquired by another firm.

**Table 1. Yearly explained variance in mean wage and standard deviation of mean wage associated with three-digit industry location for German workplaces with at least 20 employees.**

|  | Industry Explained Variance in Between Workplace Earnings Inequality | Industry Explained Variance in Within Workplace Earnings Inequalities |
|---|---|---|
| 1995 | 0.382 | 0.305 |
| 1996 | 0.414 | 0.289 |
| 1997 | 0.424 | 0.290 |
| 1998 | 0.442 | 0.342 |
| 1999 | 0.443 | 0.323 |
| 2000 | 0.459 | 0.321 |
| 2001 | 0.470 | 0.335 |
| 2002 | 0.474 | 0.289 |
| 2003 | 0.485 | 0.285 |
| 2004 | 0.490 | 0.314 |
| 2005 | 0.496 | 0.313 |
| 2006 | 0.492 | 0.305 |
| 2007 | 0.476 | 0.297 |
| 2008 | 0.483 | 0.312 |
| 2009 | 0.485 | 0.299 |
| 2010 | 0.491 | 0.295 |
| % change | 28.5% | -3.3% |

Lines represent changes in the German industrial classification system.

Occupational skill is measured with three-digit occupations ranked in terms of their mean wages converted to percentiles of the national income distribution. This measure is empirically very similar to an occupational socioeconomic status or prestige measure [29]. Others have shown across multiple countries that the consistent underlying dimension associated with occupational socioeconomic status is skill in production [30]. Previous work has used a similar measurement strategy for linked employer-employee data [4, 29]. The measurement of occupational skill as income percentile is not tautological. Only a third of individual earnings variance in Germany is between occupations [31] and the conversion to percentiles further reduces the association with individual earnings. Economy wide between workplace variance in occupational skill is predicted to rise (H3a), while occupational skill levels are hypothesized to be associated with higher (between workplaces) and increasing (within workplace) wage levels (H3c).

Occupational complexity is measured with the within workplace standard deviation of these occupational ranks. Economy wide within workplace occupational complexity is hypothesized to decline (H3b). Decreased occupational complexity is also predicted to be associated at the workplace level with increased between workplace inequality (H3d) and decreased within workplace inequality (H2b).

We calculate the workplace proportion of part-time workers to test three linked hypotheses: that establishment variance in percent part-time will increase (H4a), and that part-time labor forces will decrease between workplace full-time wages (H4b) while increasing within workplace full-time wages (H4c).

We measure workplace heterogeneity for part-time versus full-time composition as well as other categorical status attributes (see below) with the Gibbs-Martin index of heterogeneity:

$$H = 1 - \sum_{O=1}^{n} p_O^2$$

where $p_o$ is the proportion of employees in an establishment within a category and $n$ is the number of categories (e.g. two each for part-time status, sex, citizenship, and tertiary education).

## Control variables

Standard economic [4] theory leads us to expect that individual skill levels and individual skill heterogeneity should drive between and within workplace earnings inequalities, respectively. Organizational inequality [25] theory would agree, but stress the additional possibility that gender and citizenship distinctions may also be influential. We include as compositional control variables the levels and heterogeneity of tertiary degrees, age, tenure, sex and citizenship. Heteroegeneity for continuous variables (age, tenure) is measured as their workplace standard deviation. Categorical heterogeneity (education, sex and citizen composition) is measured with the Gibbs-Martin heterogeneity index as defined above. We also control for industry in final models.

## Analytic strategy

We focus first on the economy wide hypotheses and follow with hybrid multi-level regression models predicting variation in between and within workplace inequality. Hybrid multi-level regression models combine the advantage of fixed effects multi-level and cross-sectional models. The fixed effect model focuses on change, while the cross-sectional model describes stable relationships. Thus we can account for the impact of time-invariant measured and unmeasured higher level characteristics in the analysis of change, while allowing for the estimation of

time-invariant effects in the cross sectional analysis.We introduce the logic of each empirical analysis as we undertake it.

**Industry hypothesis analysis.** In our first analysis we examine H1, that between workplace earnings inequalities are increasingly tied to between industry earnings differentials. The dependent variable is workplace mean earnings. We estimate yearly cross-sectional models regressing workplace mean earnings on fixed effects for three-digit industry codes. We then focus on the level of variance explained by industry, as captured in the $R^2$ statistic. H1 predicts an increase in explained variance over time. We experimented with 2 and 4 digit industry distinctions as well, but 3 digit distinctions captured both earnings variation and trends better than the simpler and roughly equivalent to the more detailed operationalization. For the sake of comparison, we also regress workplace internal inequalities on industry with the same yearly $R^2$ focused strategy, but with no prior theoretical expectations.

We interpret Table 1 as providing strong support for H1, that between industry wage polarization is intensifying. In 1995, 38% of establishment mean wage variation was between detailed industries; by 2010 that had risen to 49%. The association of industry with workplace mean wage grew a remarkable 28.5% in only fifteen years.

We also report the parallel analysis for within workplace inequality. Within workplace inequality is more weakly associated with industry, and shows no temporal trend. Within industry workplace inequality is not becoming more similar over time.

**Descriptive analyses and economy wide hypotheses.** In Table 2 we examine three hypotheses as part of an initial descriptive analysis. We also use this table to establish if other workplace characteristics changed during this period and so are potential competing explanations for the national trends we are trying to understand. Table 2 also provides descriptive statistics for the entire panel for all variables used in the multivariate analysis that follows. The first two columns of Table 2 describe the pooled sample, the next four the first and last years of the panel, and the last two changes in the mean levels and between workplace standard deviation for all variables.

It is these last two columns that we use to examine hypotheses H3a, H3b, and H4a. We begin with these three hypotheses and then go on to describe other aspects of the changing German economy. All three hypotheses are confirmed. The German economy became more skill and part-time segregated between workplaces, while the average workplace simplified its division of labor.

We ask in hypothesis H3a if the economy wide between workplace variance in skill levels rose. In 1995, the standard deviation across workplaces in mean occupational skill was .191. This rose by 5.2% to .201 by 2010. Using a two-sample variance comparison test the null hypothesis of no change can be rejected at below the .0001 level. Between workplace skill variance rose between 1995 and 2010, confirming hypothesis H3a.

Hypothesis H3b predicts that economy wide within workplace skill complexity will decline. In 1995 the standard deviation of workplace skill complexity was .201, declining to .188 by 2010. This 6.47% drop produced a two sample *t* statistic of 10.9, rejecting the null hypothesis of no change at well below a .0001 probability level and confirming hypothesis H3b.

Our final economy wide prediction (H4a) is that the between workplace variance in part-time work will rise. In the average 1995 workplace 12.7% of jobs were part-time. This rose to 20% in 2010. Over the observation period, the proportion jobs part-time grew by 57.5%. The standard deviation across workplaces grew by 27.2%, strongly confirming hypothesis H4a. This large change in employment composition is highly statistically significant.

While the hypotheses associated with our core analyses about economy wide restructuring around industry, occupation and job precarity were all supported, other aspects of German establishments were also changing, some dramatically. Both mean tenure and the percent of

**Table 2. Establishment measures for all, first, and last years, 1995–2010.**

| | Pooled sample | | Years | | | | Mean change | |
|---|---|---|---|---|---|---|---|---|
| | 1995–2010 | | 1995 | | 2010 | | 1995–2010 | |
| | Mean | SD | Mean | SD | Mean | SD | % change, Mean | % change, SD |
| **Establishment wages** | | | | | | | | |
| Mean log earnings | 4.568 | 0.335 | 4.524 | 0.285 | 4.582 | 0.380 | 1.282 | 33.333 |
| Standard deviation of log earnings | 0.321 | 0.102 | 0.289 | 0.082 | 0.336 | 0.106 | 16.263 | 29.268 |
| **Establishment Dynamics** | | | | | | | | |
| Enters Panel after 1995 | 0.286 | 0.452 | | | | | | |
| Leaves Panel before 2010 | 0.292 | 0.456 | | | | | | |
| **Job Structure** | | | | | | | | |
| Mean occupational skill | 0.571 | 0.197 | 0.550 | 0.191 | 0.585 | 0.201 | 6.364 | 5.236 |
| Standard deviation occupational skill | 0.195 | 0.073 | 0.201 | 0.071 | 0.188 | 0.073 | -6.468 | 2.817 |
| Percent part-time jobs | 0.165 | 0.172 | 0.127 | 0.147 | 0.200 | 0.187 | 57.480 | 27.211 |
| Part-time vs. full-time heterogeneity | 0.216 | 0.163 | 0.179 | 0.158 | 0.250 | 0.158 | 39.665 | 0.000 |
| **Workforce composition** | | | | | | | | |
| Mean age | 40.326 | 4.053 | 38.631 | 3.789 | 42.081 | 4.065 | 8.931 | 7.284 |
| Mean tenure | 7.508 | 4.313 | 6.501 | 3.785 | 8.891 | 4.925 | 36.764 | 30.119 |
| % Tertiary education | 0.195 | 0.209 | 0.137 | 0.172 | 0.251 | 0.230 | 83.212 | 33.721 |
| % Men | 0.659 | 0.246 | 0.659 | 0.257 | 0.660 | 0.240 | 0.152 | -6.615 |
| % Citizen | 0.901 | 0.125 | 0.907 | 0.126 | 0.897 | 0.124 | -1.103 | -1.587 |
| **Workforce heterogenity** | | | | | | | | |
| Standard deviation of age | 9.889 | 1.418 | 10.270 | 1.418 | 10.085 | 1.474 | -1.801 | 3.949 |
| Standard deviation of tenure | 5.087 | 3.002 | 4.397 | 2.616 | 5.813 | 3.216 | 32.204 | 22.936 |
| Tertiary education heterogeneity | 0.227 | 0.167 | 0.178 | 0.157 | 0.269 | 0.163 | 51.124 | 3.822 |
| Gender heterogeneity | 0.328 | 0.148 | 0.317 | 0.151 | 0.333 | 0.146 | 5.047 | -3.311 |
| German citizen vs. non-citizen heterogeneity | 0.148 | 0.135 | 0.137 | 0.139 | 0.154 | 0.132 | 12.409 | -5.036 |
| Observations | 118397 | | 7545 | | 7134 | | | |

the workforce with tertiary degrees rose strongly, as did between workplace tenure and education segregation. The same is true, although less dramatically, for age. Human capital between establishment segregation is a plausible alternative explanation of rising between establishment inequality to job restructuring. In contrast, gender and citizen segregation between establishments declined, suggesting that these are unlikely to be explanations for between workplace inequality trends. In the multivariate models that follow we treat these compositional variables as statistical controls to examine the robustness of our core hypotheses.

## Multivariate analysis approach

We now turn to workplace level analyses of the association between establishment births, the occupational and part-time organization of work, and between and within workplace inequalities. These analyses estimate the potential influence of organizational covariates on mean logged earnings (the between workplace inequality component) and the standard deviation of logged earnings (the within workplace inequality component) for the panel of German workplaces. The vast majority of both between and within earnings inequalities are a stable function of workplaces. For mean log earnings, 96.2% of variance is at the establishment level. For the standard deviation of log earnings, the stable establishment component is smaller, but still substantial at 78.2%. Reflecting this stability, we have pursued a hybrid multi-level regression

modeling strategy. The between coefficient of the hybrid multi-level regression examines the impact of independent variables of interest on the dependent variable, taking the mean level of selected measures for each workplace across the panel [32]. The within coefficient employs a workplace fixed effect estimation strategy, focusing on change across time. Both models are estimated simultaneously with a maximum likelihood estimator. Time varying control variables influence both of these estimates, although we do not discern separate between and within components of those effects. Only establishment entry and exit variables are time invariant, and their influence is limited to the between estimate.

The random-intercept model, which provides the basis for the estimations of the hybrid models can be written as:

$$\log E_{wt} = \delta_0 + \delta X_{wt} + \beta C_w + \varphi_t + \epsilon_w + \epsilon_{wt}$$

where $log\ E_{wt}$ are the log daily workplace earnings, $X_{wt}$ indexes a vector of observed time-dependent variables illustrating workplace change over time, while $C_w$ presents a vector of observed time-independent higher-level variables at the workplace level. $\varphi_t$ are the year fixed effects and $\epsilon_w + \epsilon_{wt}$ represent the error terms.

Similar to a fixed effect strategy a hybrid model also refers to change in Y as a function of change in X over time. Fixed effect models control for time invariant omitted variable bias including, e.g., geographic location. These models can be written as:

$$\log E_{wt} = \delta_0 + \delta X_{wt} + \varphi_w + \varphi_t + \epsilon_{wt}$$

Where $\varphi_w + \varphi_t$ are workplace and year fixed effects, while $\epsilon_{wt}$ presents the error-term. The hybrid models combine the features of both multi-level regression strategies by estimating a random effects model including within effects. Technically, this is achieved by decomposing one or more variables at the lower level into a between effect ($\bar{x}_w = n_w^{-1} \sum_{t=1}^{n_w} x_{wt}$) using the mean of the variable and a within effect ($x_{wt} - \bar{x}_w$) capturing the deviation from the mean in accordance to the demeaning procedure used for fixed-effects models [32].

Hybrid models for the mean of earnings can be written as:

$$\log E_{wt} = \delta_0 + \delta X_{wt} + \beta C_w + \gamma_1 B_w + \gamma_2 W_{wt} + \varphi_t + \epsilon_w + \epsilon_{wt}$$

Where the additional term $B_w$ refers to a vector of between components and $W_{wt}$ to a vector of within components. The equation for the standard deviation of earnings models are identical. Robust (Huber-White sandwich) standard errors adjust for within workplace clustering. All observations are weighted by workplace full-time job size in order to generalize models to national inequality trends.

As always, our models are vulnerable to omitted variable bias and so can best be seen as indicators of potential causality, rather than strict causal tests. This is less the case for the within workplace fixed effect analyses, but even there changes in behavior such as in managerial practices or capital investment in new technologies are missing and potentially influential.

Perhaps the most serious missing indicator is the absence of a measure of workplace collective bargaining coverage. The rise of new, non-unionized establishments [4] and dualization in union wage bargaining [22] may permit the formation of new low earnings establishments. Assuming that establishments are born unionized (or not), this omitted variable is effectively controlled by the inclusion of establishment birth and death in the cross-sectional analyses and by design in the fixed effects specification, but it does limit our ability to comment on the magnitude of this institutional shift effect. Because unionization in Germany is strongly associated with industry and we control for industry in final models, that model is the least at risk to bias from the lack of workplace unionization measures.

More generally, with the exception of occupational skill and part-time measures, we lack information on other dynamic organizational processes such as outsourcing and subcontracting production tasks, which we expect to influence the between workplace outcomes. As such, our models can give us estimates of the impact of increased between establishment occupational skill dispersion on workplace inequality, but not the underlying mechanisms which produce changes in occupational skill composition. We rely on the initial analysis of industry linked earnings polarization and economy wide shifts in occupational and part-time structure to provide partial evidence as to the union dualization and fissuring processes. To the extent that shifts in skill and part-time composition are systematically associated with other dynamic organizational processes, estimates may be biased, though the direction of bias is unknown.

We also lack a measure of fixed versus permanent contract frequency in the workplace. In Germany, collective agreements do not allow for unequal pay for fixed-term and permanent-contract workers, although in practice temporary workers' occupational titles may be downgraded to allow for lower payments [33, 34]. Since legally it cannot influence earnings, but socially it might influence the mean occupational skill of a workplace, this source of omitted variable bias can be expected to be largely absorbed by the estimated influence of skill composition upon earnings. Thus, estimates of occupational skill effects can be seen as primarily about shifts in the division of labor, but potentially may also incorporate the impact of the rise of fixed term contracts.

We organize our multivariate analysis through four sequential models. We treat Model 1 as the baseline: it includes only a fixed effect for year. In Model 2 we introduce the variables indexing which establishments were born and died during the observation period. Model 3 adds variables indexing levels and variability in job structures. Model 4 introduces the set of employment composition variables as controls. Because industry does not appear to be a fixed trait (see Table 1), we also include it as a statistical control in Model 4. We see the contrast between Model 3 and Model 4 as providing upper and lower bound estimates of the impact of our hypothesized job structure variables on earnings inequalities.

Our fifth hypothesis predicts that within workplace inequalities would fall in both high and low wage establishments. Thus when analyzing within workplace inequality we introduce the logarithm of mean earnings as a potential explanatory variable. If a process of internal skill homogenization based on outsourcing and new firm business models is occurring at the top and the bottom of the establishment income distribution, we expect reduced inequality in these establishments. This effect of mean wage, if it exists, should be at least partly mediated by declining occupational complexity. We introduce this variable in Model 2 and observe its attenuation after occupational variables are added in Model 3.

Prior research has documented in cross-section that larger establishments pay higher wages, but also that the size premium (at least in the U.S.) has declined during this period of outsourcing and downsizing [35]. In preliminary models we explored the influence of organizational size on both mean earnings and earnings variance. A cross-sectional establishment size premium in Germany was observed in OLS models but not in the fixed effect model specification; as a result we do not include organizational size in our models.

## Modeling workplace mean wage levels and change

Table 3 reports variable and model estimates from the hybrid multilevel model for workplace mean wages. If we focus first on model estimates, reported at the bottom of the table, we confirm that the vast majority (96.1%) of variance in workplace mean wages is between establishments. Thus, change in between workplace wage inequality is primarily a function of the birth and death of establishments with different structural characteristics.

**Table 3. Hybrid models with within and between estimates in random effect regressions of mean full-time workplace earnings on organizational covariates, German workplaces with 20 or more employees, 1995–2010.**

| | (I) Baseline | | (II)Establishment births and deaths | | (III) Job structure | | (IV) Full model | |
|---|---|---|---|---|---|---|---|---|
| **Establishment births and deaths** | | | | | | | | |
| Established after 1995 | | | -0.093*** | (0.001) | -0.077*** | (0.001) | -0.021*** | (0.001) |
| Last year before 2010 | | | -0.041*** | (0.001) | 0.010*** | (0.001) | -0.004*** | (0.001) |
| **Job Structure** | | | | | | | | |
| Between-effect mean occupational skill | | | | | 1.084*** | (0.002) | 0.936*** | (0.002) |
| Within-effect mean occupational skill | | | | | 0.558*** | (0.001) | 0.537*** | (0.001) |
| Between-effect standard deviation occupational skill | | | | | 0.167*** | (0.006) | 0.086*** | (0.005) |
| Within-effect standard deviation occupational skill | | | | | -0.027*** | (0.002) | -0.040*** | (0.002) |
| Between-effect part-time | | | | | -0.106*** | (0.007) | 0.102*** | (0.006) |
| Within-effect part-time | | | | | 0.173*** | (0.001) | 0.108*** | (0.001) |
| Between-effect part-time vs. full-time | | | | | 0.179*** | (0.007) | 0.205*** | (0.006) |
| Within-effect part-time vs. full-time | | | | | -0.063*** | (0.001) | -0.026*** | (0.001) |
| **Workforce composition** | | | | | | | | |
| % Tertiary education | | | | | | | 0.240*** | (0.001) |
| Mean age | | | | | | | 0.006*** | (0.000) |
| Mean tenure | | | | | | | 0.005*** | (0.000) |
| % Men | | | | | | | 0.327*** | (0.001) |
| % Citizen | | | | | | | 0.118*** | (0.002) |
| **Workforce heterogeneity** | | | | | | | | |
| Tertiary degrees | | | | | | | -0.091*** | (0.001) |
| Standard deviation age | | | | | | | -0.007*** | (0.000) |
| Standard deviation tenure | | | | | | | 0.003*** | (0.000) |
| Gender | | | | | | | -0.016*** | (0.001) |
| German citizen vs. non-citizen | | | | | | | 0.055*** | (0.001) |
| **Year** *Reference cat.: 1995* | | | | | | | | |
| 1996 | 0.006*** | (0.000) | 0.007*** | (0.000) | 0.006*** | (0.000) | 0.003*** | (0.000) |
| 1997 | -0.004*** | (0.000) | -0.004*** | (0.000) | -0.005*** | (0.000) | -0.017*** | (0.000) |
| 1998 | 0.010*** | (0.000) | 0.010*** | (0.000) | 0.007*** | (0.000) | -0.011*** | (0.000) |
| 1999 | 0.028*** | (0.000) | 0.029*** | (0.000) | 0.024*** | (0.000) | 0.001*** | (0.000) |
| 2000 | 0.036*** | (0.000) | 0.036*** | (0.000) | 0.031*** | (0.000) | 0.006*** | (0.000) |
| 2001 | 0.039*** | (0.000) | 0.040*** | (0.000) | 0.033*** | (0.000) | 0.002*** | (0.000) |
| 2002 | 0.037*** | (0.000) | 0.037*** | (0.000) | 0.026*** | (0.000) | -0.009*** | (0.000) |
| 2003 | 0.037*** | (0.000) | 0.038*** | (0.000) | 0.027*** | (0.000) | -0.016*** | (0.000) |
| 2004 | 0.032*** | (0.000) | 0.032*** | (0.000) | 0.020*** | (0.000) | -0.027*** | (0.000) |
| 2005 | 0.023*** | (0.000) | 0.024*** | (0.000) | 0.010*** | (0.000) | -0.046*** | (0.000) |
| 2006 | 0.016*** | (0.000) | 0.017*** | (0.000) | 0.002*** | (0.000) | -0.058*** | (0.000) |
| 2007 | 0.011*** | (0.000) | 0.011*** | (0.000) | -0.004*** | (0.000) | -0.068*** | (0.000) |
| 2008 | 0.006*** | (0.000) | 0.006*** | (0.000) | -0.009*** | (0.000) | -0.073*** | (0.000) |
| 2009 | 0.012*** | (0.000) | 0.012*** | (0.000) | -0.003*** | (0.000) | -0.075*** | (0.000) |
| 2010 | 0.020*** | (0.000) | 0.021*** | (0.000) | 0.004*** | (0.000) | -0.072*** | (0.000) |
| Controls for industries (3 digit) | no | | no | | no | | yes | |
| Constant | 4.371*** | (0.001) | 4.421*** | (0.001) | 3.824*** | (0.002) | 3.417*** | (0.007) |
| Observations: Establishment-years | 118397 | | 118397 | | 118397 | | 118397 | |
| *Observations: Establishments* | *11919* | | *11919* | | *11919* | | *11919* | |
| Rho (fraction of variance at establishment level) | 0.962 | | 0.961 | | 0.941 | | 0.917 | |
| Establishment level standard deviation | 0.367 | | 0.364 | | 0.278 | | 0.220 | |

(*Continued*)

**Table 3.** (Continued)

|  | (I) Baseline | (II)Establishment births and deaths | (III) Job structure | (IV) Full model |
|---|---|---|---|---|
| Standard deviation at the lower level (establishment-years) | 0.073 | 0.073 | 0.070 | 0.066 |

Null model: Observations: 118397; Observation Workplace: 11919; Rho (fraction of variance at establishment level): 0.961; Firm-level standard deviation: 0.367; Standard deviation at the lower level (firm-years): 0074. Standard errors in parentheses.

$^*$ $p < 0.05$, $^{**}$ $p < 0.01$

$^{***}$ $p < 0.001$.

We also note from the baseline model that mean workplace inflation adjusted earnings in Germany barely changed over time, rising slowly through 2001 and declining thereafter.

To get a rough sense of the relative impact of each variable, we calculate effect sizes by multiplying estimated coefficients by the standard deviation of variables reported in column 2 of Table 2. We now turn to examine four hypotheses.

Models 2 and 3 examine hypothesis H2a, that new establishments will have lower mean wages and lower skill profiles than existing establishments. We see that new establishments have 9.3% lower mean wages than establishments that exist throughout the observation period. They also have 5.2% lower earnings than the presumably economically marginal establishments that exit the panel prior to 2010. Adding job structure variables in Model 3 attenuates the lower wage coefficient of new establishments by 17%. H2a is supported: new establishments have lower mean wages and this is in part because they are being founded with less skilled job structures.

Hypothesis H3c predicts that higher workplace skill levels will be associated with increased between and within workplace inequality. More skilled workplaces have higher mean wages, with a 1.1% rise in mean wage predicted per percentile gain in occupational skill level. The impact of changing skill levels on change in within workplace mean wage is also highly significant, but only about half as strong. Both effects are only marginally attenuated by the introduction of employment composition and industry controls. In terms of effect sizes associated with a standard deviation rather than a unit increase, a standard deviation higher workplace mean skill level is associated with 18% higher mean earnings. For the fixed effect coefficient, we find that a one standard deviation rise in skill level implies an 11% rise in that workplace's mean earnings. As expected, workplace skill levels have the strongest effect on wage levels of any variables in these two sets of models.

We did not have hypotheses about the impact of occupational complexity upon mean wage levels or change. We find, however, that higher occupational complexity is associated with high mean wages, a result consistent with classical political economy notions about the productivity advantages of detailed divisions of labor. In contrast, rising occupational complexity is associated with declining mean wages in the fixed effect estimation. However, the between effect of skill heterogeneity has an effect size (coefficient times one SD increase) of 0.01% workplace mean wage and the within effect is only -0.002%, so while statistically significant, these are not large effects.

Hypothesis H4b predicts that establishments with more part-time work will have lower full-time wage levels, net of occupational skill levels and complexity. Hypothesis H4c reflects the expectation that rising part-time work will be associated with a positive effect on within establishment full-time worker wages. Both dualization hypotheses are supported. Net of occupational skill levels and occupational complexity, more part-time workers are associated in cross-section with lower wages for full-time workers, but rising part-time labor forces

strengthen full-time workers' wages in the fixed effect estimation. The latter effect is partially mediated by employment composition. The effect sizes for both variables is moderate at 0.02, or a 2% rise in workplace mean wages for a 17% (one standard deviation) rise in part-time labor.

We lacked hypotheses on the impact of part-time versus full-time heterogeneity on mean wages. We find across workplaces that when establishments approach fifty percent part-time, full-time workers tend to be paid higher wages. However, this same process is associated with declining within workplace full-time wages. We see these results as strengthening the interpretation that dualization has short-term positive effects for full-time workers, but over the longer term undermines their bargaining power. The between effect of part-time heterogeneity has a moderate effect size of 0.03, while the within effect is only 0.01 for a one standard deviation increase.

In terms of control variables, a rising share of tertiary educated, older, longer tenure, male and citizen workers are all associated with higher average workplace earnings, even net of controls for occupational skill and skill heterogeneity. The strongest effect sizes among these variables are for sex (0.08) and education (0.05).

We did not have expectations linking heterogeneity measures to mean earnings. All of those effect sizes are relatively modest, below a 2% change in workplace mean wage. Looking across variables, it is clear that the strongest contribution to between workplace mean earnings and to within workplace mean earnings change derives from the skill level of the organization. Surprisingly, the next strongest effect size derives from the sex composition of the workplace, followed by the educational level of the workforce, and then the new establishment effect. All other covariates display relatively weak influences on earnings levels and change in these models

**Within workplace inequality.** In Table 4 we explore the correlates of within workplace earnings variation. Focusing first on the baseline model Rho statistic at the bottom of the table, we see that about three-quarters of variance in within workplace inequality is a stable attribute of establishments. The time trend reported in the baseline model confirms that average within workplace inequality has been rising across the study period. We next examine three formal hypotheses relating to within workplace inequalities.

The first hypothesis (H2b) predicts that new establishments will have lower internal inequalities and lower skill complexity than existing establishments. Net of workplace mean earnings, and inconsistent with hypothesis H2b, new establishments have the same internal earnings inequalities as existing workplaces. After controls for job structure, industry, and status composition, new establishments have significantly higher inequalities than existing workplaces. In contrast, workplaces that exit the panel had lower internal inequality, although this seems to be primarily a function of job structure.

We also hypothesized that (H3d) declining workplace skill complexity would be associated with decreased within workplace inequality. For both between and within workplace models of internal earnings inequalities we observe strong positive relationships between skill complexity and the level and change in within workplace inequalities. Since the economy wide trend is toward declining workplace occupational complexity, hypothesis H3d is supported. The effect sizes are somewhat larger than most others for within workplace inequalities at .03 for between and .02 for within estimates.

Our final set of hypotheses focus on the relationship between workplace wage levels and internal inequalities. Hypothesis H5a states that both rising and declining workplace earnings will be associated with smaller internal earnings inequalities, while hypothesis H5b predicts that high wage establishments will tend to have the lowest levels of internal inequalities. We produce marginal effect plots (Fig 1), adjusting for model covariates to examine these non-linear patterns. We focus on Model 3 estimates, but the full model estimates are quite similar.

**Table 4. Hybrid models with within and between estimates in random effect regressions of standard deviation of full-time workplace earnings on organizational covariates, German workplaces with 20 or more employees, 1995–2010.**

| | (I) Baseline | | (II) Establishment births and deaths | | (III) Job structure | | (IV) Full model | |
|---|---|---|---|---|---|---|---|---|
| **Establishment births and deaths** | | | | | | | | |
| Established after 1995 | | | -0.000 | (0.000) | 0.001** | (0.000) | 0.006*** | (0.000) |
| Last year before 2010 | | | -0.012*** | (0.000) | -0.004*** | (0.000) | -0.001* | (0.000) |
| **Establishment wages** | | | | | | | | |
| Between-effect of mean log earnings | | | -1.039*** | (0.006) | -1.238*** | (0.006) | -0.750*** | (0.005) |
| Between-effect of mean log earnings squared | | | 0.128*** | (0.001) | 0.145*** | (0.001) | 0.091*** | (0.001) |
| Within-effect of mean log earnings | | | -1.186*** | (0.005) | -1.334*** | (0.005) | -1.317*** | (0.005) |
| Within-effect of mean log earnings squared | | | 0.127*** | (0.001) | 0.141*** | (0.001) | 0.142*** | (0.001) |
| **Traditional productivity measurements** | | | | | | | | |
| Between-effect mean occupational skill | | | | | 0.144*** | (0.001) | 0.103*** | (0.001) |
| Within-effect mean occupational skill | | | | | 0.189*** | (0.001) | 0.134*** | (0.001) |
| Between-effect standard deviation occupational skill | | | | | 0.335*** | (0.002) | 0.235*** | (0.002) |
| Within-effect standard deviation occupational skill | | | | | 0.297*** | (0.001) | 0.252*** | (0.001) |
| Mean age | | | | | | | -0.000*** | (0.000) |
| Mean tenure | | | | | | | -0.003*** | (0.000) |
| Standard deviation age | | | | | | | 0.005*** | (0.000) |
| Standard deviation tenure | | | | | | | 0.005*** | (0.000) |
| **Workforce composition** | | | | | | | | |
| Between-effect part-time | | | | | -0.059*** | (0.002) | -0.015*** | (0.002) |
| Within-effect part-time | | | | | -0.014*** | (0.001) | -0.004*** | (0.001) |
| Tertiary education | | | | | | | 0.039*** | (0.001) |
| Men | | | | | | | -0.030*** | (0.000) |
| Citizen | | | | | | | -0.027*** | (0.001) |
| **Workforce heterogeneity** | | | | | | | | |
| Between-effect part-time vs. full-time | | | | | 0.110*** | (0.002) | 0.004 | (0.002) |
| Within-effect part-time vs. full-time | | | | | -0.006*** | (0.001) | -0.014*** | (0.001) |
| Tertiary degrees | | | | | | | 0.052*** | (0.001) |
| Gender | | | | | | | 0.130*** | (0.001) |
| German citizen vs. non-citizen | | | | | | | 0.020*** | (0.001) |
| **Years** *Reference cat.: 1995* | | | | | | | | |
| 1996 | -0.003*** | (0.000) | -0.003*** | (0.000) | -0.002*** | (0.000) | -0.003*** | (0.000) |
| 1997 | -0.001*** | (0.000) | -0.001*** | (0.000) | -0.001*** | (0.000) | -0.001*** | (0.000) |
| 1998 | 0.014*** | (0.000) | 0.013*** | (0.000) | 0.013*** | (0.000) | 0.012*** | (0.000) |
| 1999 | 0.015*** | (0.000) | 0.015*** | (0.000) | 0.014*** | (0.000) | 0.012*** | (0.000) |
| 2000 | 0.022*** | (0.000) | 0.021*** | (0.000) | 0.021*** | (0.000) | 0.016*** | (0.000) |
| 2001 | 0.025*** | (0.000) | 0.025*** | (0.000) | 0.025*** | (0.000) | 0.021*** | (0.000) |
| 2002 | 0.041*** | (0.000) | 0.039*** | (0.000) | 0.039*** | (0.000) | 0.035*** | (0.000) |
| 2003 | 0.024*** | (0.000) | 0.023*** | (0.000) | 0.023*** | (0.000) | 0.019*** | (0.000) |
| 2004 | 0.025*** | (0.000) | 0.023*** | (0.000) | 0.022*** | (0.000) | 0.018*** | (0.000) |
| 2005 | 0.028*** | (0.000) | 0.026*** | (0.000) | 0.025*** | (0.000) | 0.022*** | (0.000) |
| 2006 | 0.031*** | (0.000) | 0.028*** | (0.000) | 0.026*** | (0.000) | 0.023*** | (0.000) |
| 2007 | 0.034*** | (0.000) | 0.030*** | (0.000) | 0.029*** | (0.000) | 0.024*** | (0.000) |
| 2008 | 0.035*** | (0.000) | 0.031*** | (0.000) | 0.029*** | (0.000) | 0.021*** | (0.000) |
| 2009 | 0.033*** | (0.000) | 0.029*** | (0.000) | 0.028*** | (0.000) | 0.020*** | (0.000) |
| 2010 | 0.034*** | (0.000) | 0.030*** | (0.000) | 0.029*** | (0.000) | 0.020*** | (0.000) |
| Controls for industries (3 digit) | no | | no | | no | | yes | |

*(Continued)*

**Table 4.** (Continued)

|  | (I) Baseline | | (II) Establishment births and deaths | | (III) Job structure | | (IV) Full model | |
|---|---|---|---|---|---|---|---|---|
| Constant | 0.279*** | (0.000) | 2.362*** | (0.014) | 2.768*** | (0.013) | 1.692*** | (0.012) |
| Observations: Establishment-years | 118397 | | 118397 | | 118397 | | 118397 | |
| *Observations: Establishments* | 11919 | | 11919 | | 11919 | | 11919 | |
| Rho (fraction of variance at establishment level) | 0.782 | | 0.752 | | 0.729 | | 0.635 | |
| Establishment level standard deviation | 0.098 | | 0.089 | | 0.082 | | 0.065 | |
| Standard deviation at the lower level (establishment-years) | 0.052 | | 0.051 | | 0.050 | | 0.049 | |

Null model: Observations: 118397; Observation Workplace: 11919; Rho (fraction of variance at establishment level): 0.780; Firm-level standard deviation: 0.100; Standard deviation at the lower level (firm-years): 0053. Standard errors in parentheses.

* $p < 0.05$

** $p < 0.01$

*** $p < 0.001$.

Both hypotheses must be rejected. Workplaces with rising and falling wages display strongly rising internal inequalities, although the magnitude of this effect is stronger for low wage workplaces. The pattern is similar in cross-section, although the pattern is more symmetrical. Within workplace wage homogeneity is most likely to be found in establishments with little or no shifts in their mean wages and in the middle of the establishment wage distribution. Low wage and high wage establishments are better characterized as within workplace inequality generators.

As expected, the various status heterogeneity measures are also associated with higher within workplace inequality. Looking across these measures of internal heterogeneity, gender and occupational complexity have the largest effect sizes, followed closely by both education and tenure. While the results for education and tenure are consistent with a standard human capital model, the results taken together suggest that German within workplace inequality is not simply a result of variance in individual human capital levels, but that occupational divisions of labor and gender are both at least as influential. The expectation that occupational skill variance would be the primary driver of within workplace inequality is not confirmed. At least in these dynamic models, education, tenure, gender, and occupation all appear to be

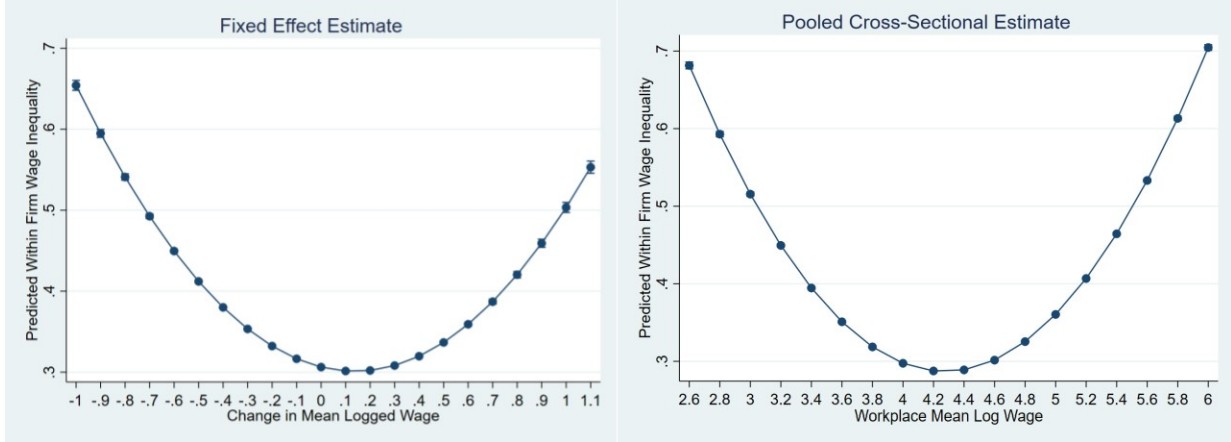

**Fig 1. Predicted levels of within workplace wage inequality as a function of workplace mean wage levels and change.**

roughly equivalent drivers of German internal workplace inequality dynamics in the study period.

## Discussion

Following the increasing political and scholarly interest in rising earnings inequalities, we focus on Germany, a nation with particularly steep increases in both between and within workplace inequality. We depart from prior work in three fundamental ways. First, we foreground workplaces as the sites of income pooling and distribution. Second, we connect the empirical recognition of rising between establishment inequality to the organizational fissuring and dualization literatures on the reconfiguration of larger firms through the creation of dualized job structures and the externalization of labor costs associated with outsourcing, subcontracting, and the rising market power of firms in oligopolistic industries. Finally, we directly model organizational characteristics and dynamics on shifts in both between and within workplace inequalities.

Confirming our first hypothesis, we find rapidly rising between industry polarization in mean workplace earnings. Outsourcing tasks from high to low wage firms, increased market power of dominant firms, and collective bargaining dualization are the theorized causes of this predicted industry linked earnings polarization. While none of these mechanisms are directly observed, the industry analysis helps to establish the degree to which they are potentially reasonable explanations. This industry approach does not capture the consequences of some forms of organizational reconfiguration including within industry and global outsourcing, or the externalization of labor via independent contractors.

We also directly estimate the association between a series of organizational factors and both between and within workplace inequalities. Reflecting that most of the variability in between and within workplace inequality is a stable establishment attribute, we model these impacts in a hybrid hierarchical linear modeling framework. In hybrid models a conventional fixed effect approach models change within establishments, while a pooled panel cross-sectional approach models the between establishment variance. More than 90% of between and 75% of within workplace inequality is captured in the pooled panel analyses.

Confirming hypothesis H2a, we find that between workplace inequality is strongly tied to the birth of new low wage workplaces and that these workplaces tend to have lower skill levels and lower skill complexity than more stable establishments. There is, however, no evidence that new establishments have lower internal inequality, and the sign actually switches to positive in the fully controlled model. Thus, hypothesis (H2b) must be rejected in part: new establishments do have lower skill complexity, but they have higher, not lower, internal inequalities.

Not surprisingly, we also find that occupational skill levels are strong predictors of workplace mean wage and occupational complexity of internal inequalities. As we saw Table 2, between workplace occupational skill variability grew by 5.2% and within workplace occupational complexity declined by 6.5% in this period. We see these results as quite consistent with the more general fissuring account of work reconfigurations. Workplaces are becoming internally more skill homogenous and more skill polarized from one another, and these trends are tied to both the birth of new establishments and industry earnings polarization.

We confirm the speculation that the rise of part-time jobs might be undermining the bargaining power of full-time labor [4], but demonstrate that this is primarily a between establishment phenomena. Within establishments, rising part-time labor is associated with an increase in the wages of full-time labor, a result consistent with the dualization literature [21, 22]. Consistently, research shows that in Germany the use of temporary workers increases the job security of permanent full-time workers [33]. Thus dualization appears to have two faces. Within

establishments, part-time and perhaps temporary contract work increases the wages and stability of the remaining full-time workers. At the same time, the bargaining power of all workers is reduced at the establishment level.

The hypotheses linking establishment mean wage and internal inequalities were strongly rejected. Low and high wage establishments, as well as establishments that both reduce and raise wage levels over time, are markedly more internally unequal than establishments in the middle of the wage distribution and those that exhibit stable wage levels. From a mechanical point of view, one might expect that high wage establishments should have higher inequalities simply because the social space to make distinctions rises as an organization moves away from the floor of minimum wages. That inequality is also high in low wage establishments suggests a high level of exploitation in those establishments. This empirical observation certainly deserves further exploration, both in Germany and elsewhere.

Past research found that educational sorting was not associated with rising between establishment inequalities (net of occupational shifts) for West German men [4]. In contrast, we find that shifts in the educational composition of workplaces are associated with both rising between and within workplace inequalities. Although effect sizes were consistently smaller than occupational skill effects, we saw in Table 2 that the degree of educational segregation between establishments grew by 30%, substantially faster than between workplace skill segregation. Since occupational skill and educational sorting are the demand and supply side of the same coin, we see these results as consistent with the fissuring expectations, but we cannot from these analyses adjudicate between educational or occupational structure segregation as the primary driver of rising between workplace inequality.

An additional strong empirical result has to do with the gender composition of the workforce. Rising female employment is associated with lower mean workplace earnings, net of individual and occupational skill levels. But since female employment levels do not change much over the observation period and between workplace gender segregation actually declines, this cannot be a major driver of rising inequality between establishments. In contrast to gender, citizenship composition, while statistically significant, had very small effect sizes. This is consistent with prior work that shows that citizenship is a weak status distinction in German workplaces [36].

## Conclusions

The fundamental contribution of this paper is to redirect research on growing national inequalities towards observing the firm level processes that generate market income distributions. About two-thirds of Germany's rapid growth in income inequality was produced by increased between workplace earnings inequalities. This pointedly underlines the importance of organizational wage polarization as an emergent inequality generating phenomena. Our past research [6] has shown that this pattern is not confined to Germany, but exists in twelve of fourteen high-income countries examined.

This research confirms that in Germany that a variety of organizational factors–industry wage polarization, the birth of new low wage workplaces, occupational restructuring, and the rise of labor market dualization–contribute to rising inequality both between and within workplaces in Germany. We see the results in this paper as offering confirmation of predictions from literatures on employment [4, 22] and union bargaining [24] dualization, and on workplace fissuring [7]. While there is good evidence that German establishments are becoming more skill homogenous and that this drives between workplace inequality, the suspicion that this leads to lower internal inequalities must be rejected. Internal inequalities are strongly exaggerated in low and high wage establishments and in establishments that restructure toward high and low wage divisions of labor.

We also found two sets of results that led us to reject hypotheses. Inequality is higher and rising in both declining and rising wage workplaces. We also find that new firms tend to have higher internal inequalities net of internal divisions of labor, industry, and status composition. Together these two sets of results suggest that dynamic tendencies are toward more internal firm inequalities. Clearly change, whether the birth of new firms or changes in firm pay levels, is empirically consistent with rising internal workplace inequalities. What the mechanisms are behind these dynamics is not at all apparent and deserves further theorization and empirical investigation.

We have only scratched the surface of these processes. Future research should further explore both organizational and institutional mechanisms, including those that we do not observe in this paper, such as the role of firm market power in both generating high wage jobs and creating a fissured economy. The role of outsourcing, independent contracting, global supply chains, franchising, and other strategies to externalize production while capturing economic profits should be examined as well. Why both high and low earnings workplaces are also high inequality is a finding in search of an explanation.

We do not expect any universal pattern to be discovered in future research, but rather that more proximate mechanisms will be more or less prominent in different institutional and historical settings [26]. More generally, however, we see these two sets of global mechanisms as linked. Declining organized worker power makes it easier for firms to pursue externalization of labor, while that strategy further weakens collective labor organization [see 37 for this argument applied to Germany].

This paper focuses on the variance in logged earnings, analyzed separately as between and within workplace components. While this is standard in the literature on firm wage effects [4–6, 8–10], it prioritizes inequalities around and below the mean. Future research might pursue a more distributional approach, either calculating wage ratios (e.g. 90/10, 90/50, 50/10) or moving to a quantile regression framework.

At a more theoretical level, this paper confirms the utility of exploring organizational wage setting processes, a focus that is now emerging in economics, sociology, and organizational sciences [1–3]. The paper also shows that a singular focus on traditional human capital explanations of both workplace inequality levels and change is much too narrow. Explanations need to be developed within a more organizationally focused analyses of the division of labor, which admit the potential impact of gender and other status processes, the birth and death of establishments with different inequality profiles, and the institutional contexts in which they operate [23–26].

## Supporting information

**S1 File. Imputation strategy for top coded income.**
(DOCX)

## Acknowledgments

Peter Jacobebbinghaus, then at the Institute for Employment Research (IAB) in Nuremberg, was central to the development of both sampling and imputation methods used in this paper. The paper has been improved by comments at seminars at the Wharton School of Business, Columbia University, Sociology, the University of Texas, Austin Population Center, Max Po and Sciences Po, Paris, and le Centre de Recherche en Economie et Statistique, Paris. The OrgNets seminar at the University of Massachusetts, Amherst provided a particularly useful discussion. Special thanks to Jasmine Kerrisey, Anthony Rainey, Eunmi Mun, Anthony Paik, Matt Mendoza, and Sander Wagner. Correspondence to tomaskovic-devey@soc.umass.edu.

## Author Contributions

**Conceptualization:** Donald Tomaskovic-Devey, Silvia Maja Melzer.

**Data curation:** Silvia Maja Melzer.

**Formal analysis:** Silvia Maja Melzer.

**Funding acquisition:** Donald Tomaskovic-Devey.

**Methodology:** Donald Tomaskovic-Devey, Silvia Maja Melzer.

**Project administration:** Donald Tomaskovic-Devey.

**Resources:** Silvia Maja Melzer.

**Visualization:** Silvia Maja Melzer.

**Writing – original draft:** Donald Tomaskovic-Devey.

**Writing – review & editing:** Donald Tomaskovic-Devey, Silvia Maja Melzer.

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
