## [Decision Letter · Decision Letter 0]

26 Mar 2020

PONE-D-20-03141

The Organizational Production of Earnings Inequalities, Germany 1995-2010

PLOS ONE

Dear Dr. Tomaskovic-Devey,

Thank you for submitting your manuscript to PLOS ONE. After careful consideration, we feel that it has merit but does not fully meet PLOS ONE’s publication criteria as it currently stands. Therefore, we invite you to submit a revised version of the manuscript that addresses the points raised during the review process.

We have received detailed reports from two expert reviewers. As you'll see in their reports, their recommendations split: Reviewer 2 recommends rejection, while Reviewer 1 is more positive. After carefully reading your paper and the reports of the reviewers, I decided to give you a chance to address the reviewers' comments and criticism. I would like to emphasize that invitation for resubmission at this stage does not guarantee eventual publication.

We would appreciate receiving your revised manuscript by May 10 2020 11:59PM. To enhance the reproducibility of your results, we recommend that if applicable you deposit your laboratory protocols in protocols.io, where a protocol can be assigned its own identifier (DOI) such that it can be cited independently in the future. For instructions see: http://journals.plos.org/plosone/s/submission-guidelines#loc-laboratory-protocols

We look forward to receiving your revised manuscript.

Kind regards,

Semih Tumen, PhD

Academic Editor

PLOS ONE

Journal Requirements:

2. Please clarify if the data were analyzed anonymously and provide further detail on how data were accessed and when and how the necessary permissions were obtained

Reviewers' comments:

Reviewer's Responses to Questions

**Comments to the Author**

1. Is the manuscript technically sound, and do the data support the conclusions?

Reviewer #1: Partly

Reviewer #2: No

2. Has the statistical analysis been performed appropriately and rigorously? 

Reviewer #1: Yes

Reviewer #2: No

3. Have the authors made all data underlying the findings in their manuscript fully available?

Reviewer #1: Yes

Reviewer #2: Yes

4. Is the manuscript presented in an intelligible fashion and written in standard English?

Reviewer #1: Yes

Reviewer #2: No

5. Review Comments to the Author

Reviewer #1: Review: The Organization Production of Earnings Inequalities, Germany 1995-2010

PLOS ONE

This study explores between- and within-firm components of rising income inequality in Germany. The data used are perfect for the task, and paper engages with appropriate literature in the framing and motivation of the study. I believe this will be a valuable contribution.

There are a few issues I believe the authors should consider in revisions.

1) I’m not confident that model 1 in Table 2 is actually predicting between-firm inequality via the dependent variable of mean logged workplace earnings. It seems to me that it’s predicting change in mean earnings within-organizations. This is because the models include a fixed effect for workplaces, so remaining variation in the dependent variable is that that pertaining to within- and not between-workplaces. It is not uncommon (albeit imperfect) for research to use mean wages as a measure of productivity, but here it is used as a measure of between-organization inequality. This is not inherently problematic, but given that the estimation strategy is focused on within-variation rather than between-variance, I think it may be missing the mark.

I am open to the authors’ arguments on why this approach is justifiable, but I also want to provide a couple suggestions that they may consider. Since between-firm income inequality is largely contingent on having stratified local labor markets, the researchers may wish to leverage geospatial variation to construct a measure of between-firm wage variability (sd) measured across local labor markets (metros, commuting zones, etc…) and use this as their dependent variable. This more closely measures between-firm variation in wages, using the same logic that the authors presently employ in their measure of within-firm variation. In essence, focusing on within-geography between-firm variability more closely captures the dynamic the authors describe on page 28, where the manufacturing firm outsources the cleaning and cafeteria work to other firms, presumably located in the same local labor market.

Another potential solution is to better leverage the hierarchical structure of the data with HLM or, to retain some of the fixed effects, a within-between hybrid model. Predicting individual-level workers’ logged wages and decompressing the error term into individual-level and work-place variance is another way to more closely observe the role of between workplace inequality. Examining how the addition of focal covariates affects the between-workplace error variance could be used to test the related hypotheses.

Whichever strategy the authors’ decide (particularly if they choose to continue with the present approach), I advise them to add more justification as to how, specifically, their dependent variable measures between-organization inequality.

2) The testing of H1 with Figure 2 was a bit vague. It is unclear if the model is specified similarly as in the Equation on page 18, or if there are no covariates in the model. Relatedly, Figure 2 is difficult to interpret because it appears that the industry explained between-firm variance is greater than the total between-firm variance. This is because the y-axis has a different meaning for each measure. I suggest removing Figure 2 and including Appendix Table 1 in the main text. Also, I believe adding more detail to the methods’ section on how these are estimated would be useful. Presently, stating “yearly fixed effects for three-digit industry codes” is unclear and could be better stated along the lines of “we predicted workplace mean logged earnings with fixed effects for industry using independent models across each year of data.”

3) Relatedly to point 2, I think the authors might be inferring a lot from the industry fixed effects than what is actually possible given the data. Between-industry fixed effects on firm wages may be related to bargaining and unions, but it could also reflect varying sector productivity and a host of other unobservables. One way to better isolate the effects of declining worker power is to focus on comparative industries, one where unions have declined and another where they’ve remained strong. The authors state on page 7 that low-wage service sectors and export-oriented manufacturing are some respective examples. Perhaps comparing wages between workplaces in retail and tech manufacturing would be advantageous as a valuable robustness test. Or, aligning with the example on page 28, food service and export-intensive manufacturing.

4) starting the introduction with Figure 1 left me with more questions about how the figure was calculated than it did motivate the study. Positioning the paper with the literature does a fine job of motivating the study, I don’t think Figure 1 is necessary (perhaps after introducing the data and methods). If the authors feel strongly otherwise, including extensive footnotes to the figure or directing readers to the methods section would be useful. I prefer the former.

Minor issues:

There are a couple of spelling/grammar errors throughout.

Table 2 could use more descriptive labels for r2 and r2_a.

The authors may wish to consider fixed effects for region to control for economic spatial agglomeration that affects firm productivity.

Not totally necessary to revise, but if the authors intend for the article to be accessible to non-academics (a benefit of publishing with an open-access journal) they may wish to revise the writing to remove jargon.

Reviewer #2: Many thanks for the possibility to review the paper The Organizational Production of Earnings Inequalities, Germany 1995-2010. In it, the authors investigate how institutional changes, namely the outsourcing of low and middle income occupations explains the observed increase in within- and between-workplace earnings inequality in Germany. As the authors claim, their contribution consists of evaluating how workplaces generate rising earnings inequalities, assessing the relative importance of a series of organizational factors. The paper makes strong arguments and claims but it largely fails to address them adequately. In this regard, I have several concerns:

1. A mismatch between theory and analysis. The authors present human capital and relational inequality as two theories to explain the observed patterns of rising earnings inequality. While the presentation is theoretically sound, they fail to demonstrate how to assess the claims made by the theories in the empirical section. While this is certainly also due to the data used, they nevertheless make little effort to attribute observed associations to the different theoretical arguments. This is especially prominent in the conclusion section, where linkages to the theoretical claims are largely absent. Likewise, there are many instances where it is unclear how the two theoretical approaches can be distinguished in the analyses (e.g., in lines 200-205: Rising variance in the occupational division of labor could also be explained by changing human capital differences between low- and high-skilled workers). More substantially, from the outsourcing argument follows, as the authors state in lines 565ff., that a lower within-workplace inequality and a larger between-workplace inequality should be observed which is, as they show, not the case.

2. As the authors state themselves, the core argument, the outsourcing of low-skilled, low-payed workers cannot be assessed directly with the data. While the authors present compelling arguments how they should find this pattern in their data, a more critically discussion of this limitation is needed in the conclusion section.

3. There are several methodological aspects that need adjustments. While some limitations regarding potential omitted variable bias are directly addressed by the authors, others are not. For example, the authors state that the fixed effects model should control for the likely effects of the rise of new, non-unionized firms in the time period under study. This is most certainly not the case if this is indeed a time varying, unobserved influence! Regarding sample selection: The lack of self-employed and ceiling the inclusion probability at 0.3 (although I quite understand the reason for this) might hamper the generalization of the results. One could argue that especially large firms outsource low-skilled jobs. At the same time, one would at least need some information about the extent of self-employment in Germany and to discuss these limitations again critically in the conclusion section. Although I understand why you want to impute missing values, you should adjust for the additional uncertainty when doing so, either by means of a probabilistic imputation method or by adding additional uncertainty, for example by random draws from the corresponding error distribution. Likewise, controlling for missing information in some status characteristics by means of an indicator variable likely also biases results (see, for example, Allison 2001) and setting the corresponding correlations to 0 likely influences results.

Minor comments:

• Part-time effect: Is it possible that this (unexpected) effect could be explained by labor market segments? That is, that part-time is especially prominent in some sectors, where a large share of the employees works part-time, low-payed jobs?

• The notation in the fixed effects model and its discussion in lines 387ff. is inconsistent (I guess there should only be two subscripts, w and t, and j equals w).

• Maybe provide some information when you first introduce the part-time hypotheses why you do not use fixed-term contracts (as you do in a later part of the manuscript).

• Maybe also briefly justify the use of (log) variances as the main measures rather than something like gini to assess inequality.

• Proofread the whole thing!

All in all, I am convinced that this manuscript has great potential but at the current state I do not perceive it as ready for publication.

6. PLOS authors have the option to publish the peer review history of their article (what does this mean?). If published, this will include your full peer review and any attached files.

Reviewer #1: No

Reviewer #2: No

---

## [Author Response · Author response to Decision Letter 0]

11 Jun 2020

Dear Reviewers

Thank you for a very insightful pair of reviews. We have changed the paper in multiple ways as a result. We outline these in point-by-point responses to the reviewers. Not in response to the reviewers, but to simplify the exposition, we dropped the analysis of segregation effects on internal inequalities. As you may recall from the last version these effects were quite weak, and added a great deal of complexity to an already too complex analysis. 

Reviewer 1

1. The reviewer points out that the FE models really focus on within workplace change and suggest we move into a hybrid HLM framework. This turned out to be a very productive suggestion, as doing so revealed that 90% of the workplace variance in mean wages was a stable between workplace characteristic. It was 75% for within workplace inequality. This led us to both a somewhat different interpretation of the overall statistical exercise and to add a direct modeling of the impact of the birth and death of workplaces on the between and within components. We found that new firms pay mean wages 9% lower than more long lived firms, and 4% lower than firms that exit the panel. The change to a hybrid model also led to a series of qualifications in the hypotheses as to whether they applied primarily to between or within workplace inequalities, which was a very useful exercise.

2. We now include more detail on how the industry fixed effect analysis tied to hypothesis 1 is estimated, and have dropped the figure in favor of tabular presentation. We have added a parallel analysis of within workplace inequality here as well.

3. The reviewer also wondered if rather than reflecting the causal force of industry that the observed between industry wage polarization might reflect something correlated with industry. We agree and now make clearer that we see two processes underlying this result: increased market power associated with high wage firms and the outsourcing of lower paid jobs to new low wage firms in different industries. Neither market power or outsourcing are observed, but rather are the mechanisms pointed to in the literature. We make this distinction between observed effects and unobserved mechanisms clearer throughout the manuscript. We appreciate the reviewer’s suggestion to design a comparative industry study and will implement it in a future paper.

4. Reviewer 1 points out that the literature review adequately motivates study and so we do not need Figure 1. We have dropped that figure.

5. We now make clear that we use the variance in logged earnings because it has become standard in this literature, but in the conclusion suggest that future research might use other measures sensitive to particular positions in the income distribution like 90/10 ratios or quantile regression.

Reviewer 2

1. Points out that we do not distinguish between human capital and relational inequality theories, or connect the empirical results back to these theories. We think the last version of the paper was confusing in this regard. Our core use of HC and RIT is to build a reasonable set of control variables as a robustness test of the more basic proposition that changes in firm job structure are producing rising inequalities. We have reorganized the paper to reflect this distinction more clearly. It is also the case that we do not see HC and RIT as competing, but rather complementary, descriptions of plausible processes in workplace level wage setting. We do comment on the importance of both educational levels and gender composition for explaining both between and within workplace inequalities in the discussion section, but these are not our primary focus.

2. The conclusion now includes a more critical discussion of the limitations of our analysis and the types of research that would be useful going forward. 

3. Reviewer 2 points out that we do not have a direct measure of new non-union firms and that this is a prominent alternative or complementary explanation of rising between workplace inequality trends. While we do not know the union status of workplaces, we have now included an indicator of new firms (those that enter our panel after 1995). This does not change other results, but is a result in its own right – new firms pay lower wages than either continuing firms or those that exist the panel. 

4. Reviewer 2 wonders if the under sampling of large firms for confidentiality reasons leads us to under observe potential outsourcing firms. All of our analyses use sampling weights to adjust the final sampling estimates to correct for this initial under sampling, so the answer is probably not.

5. Reviewer 2 wants to know more about the self-employed in German, who are excluded from our analyses. We have added descriptive material on the size and earnings of the self-employed in Germany in our discussion of sample exclusions.

6. The implications of part-time as a potential source of power differentials between and within firms is now much clearer in the paper. This reflects are move to hybrid models, which reveals that more part-time labor undermines full-time wages between firms, while supporting wage gains for full-time labor within firms. This result is novel and an important adjudication between the dualization and worker power predictions. We agree that the growth in part-time jobs is also associated with industry and firm segmentation, and this is one of the reasons why we control for industry in the final model and hypothesize the dualization effect within workplaces.

7. Reviewer 2 points out that union membership is not time stable. We now explain that we see it as stably associated with firms. We see the addition of a new set of indicators for establishment births and deaths as partial controls for omitted variable bias. We have also redesigned the analysis to add industry as a control variable, which should perform a similar function.

8. Smaller comments

a. We now point our earlier in the manuscript that we do not have an indicator of fixed-term contracts.

b. We use variance in logged earnings as our main inequality measure because it is standard in this literature. We do not pursue an analysis of alternative measures, such as the gini. We do point out in the conclusion that such analyses might be informative, although we stress distributional analyses such as the 90/10 ratio or quantile regression approaches.

---

## [Decision Letter · Decision Letter 1]

23 Jul 2020

PONE-D-20-03141R1

The Organizational Production of Earnings Inequalities, Germany 1995-2010

PLOS ONE

Dear Dr. Tomaskovic-Devey,

Thank you for submitting your manuscript to PLOS ONE. After careful consideration, we feel that it has merit but does not fully meet PLOS ONE’s publication criteria as it currently stands. Therefore, we invite you to submit a revised version of the manuscript that addresses the points raised during the review process.

We look forward to receiving your revised manuscript.

Kind regards,

Semih Tumen, PhD

Academic Editor

PLOS ONE

Reviewers' comments:

Reviewer's Responses to Questions

**Comments to the Author**

1. If the authors have adequately addressed your comments raised in a previous round of review and you feel that this manuscript is now acceptable for publication, you may indicate that here to bypass the “Comments to the Author” section, enter your conflict of interest statement in the “Confidential to Editor” section, and submit your "Accept" recommendation.

Reviewer #1: All comments have been addressed

Reviewer #2: (No Response)

2. Is the manuscript technically sound, and do the data support the conclusions?

Reviewer #1: Yes

Reviewer #2: Yes

3. Has the statistical analysis been performed appropriately and rigorously? 

Reviewer #1: Yes

Reviewer #2: Yes

4. Have the authors made all data underlying the findings in their manuscript fully available?

Reviewer #1: Yes

Reviewer #2: Yes

5. Is the manuscript presented in an intelligible fashion and written in standard English?

Reviewer #1: Yes

Reviewer #2: Yes

6. Review Comments to the Author

Reviewer #1: The revised paper more clearly illustrates the factors contributing to growing within- and between- workplace inequality. I particularly enjoyed the use of hybrid modelling. I believe this method is underused in the social sciences, and the authors have done a great job discussing the principles behind it and reporting the results in a clear way.

In addition, I found the findings regarding skill complexity, mean wages, and gender to be quite interesting. The surprising findings regarding the non-linear relationship of mean-wages to workplace inequality suggests that any changes in mean wages essentially increase inequality – likely representing the tendency for wage premiums to be experienced by high-earners (during growth) and wage cuts to be experienced by low-earners (during decline). Theoretically speaking, this is likely evident of the decline of labor power, which the authors cannot empirically demonstrate but speak to in their discussion.

There is certainly more to glean from the relationship of skill complexity and gender to workplace inequality. I encourage the authors to pursue this in their future research. As previous studies have focused on the deskilling of feminized occupations (e.g. Levanon, England, and Allison 2009), the authors have found some preliminary evidence of similar dynamics going on at the firm level. This is, of course, beyond the scope of the present paper. But I am excited for its future possibilities.

My only remaining recommendation for the present study is to revise the abstract. Stating that the authors “develop a set of hypotheses” and “model these hypothesized processes” without stating those hypotheses makes it difficult to really know the aims of the study by just reading the abstract (which, for many scholars, will be the only thing they read unless their interest is piqued). Some of the language in the introduction could be incorporated into the abstract to replace the focus on “hypotheses.” Something along the lines of “We explore a set of organizational explanations for rising between and within workplace inequality focusing on the role of employment dualization, skill segregation/complexity, and firm fissuring.” Then briefly state the method used and proceed to the key findings, “We find that rising …”

In addition, I think a more descriptive statement than “establishment level internal simplifications” could be used on page 4. If this is referring to occupational skill, than stating that more directly would be helpful.

Reviewer #2: Many thanks for the possibility to re-review the paper The Organizational Production of Earnings Inequalities, Germany 1995-2010. I congratulate the authors for improving their manuscript greatly since the original submission and I am convinced that it will make an important contribution. The manuscript follows a coherent logic, clearly outlines its aims and assumptions, and empirically tests them with adequate methods. I am especially grateful for the authors’ critical discussion of their main assumptions and potential selection and omitted variable issues as well as the descriptive analyses. This said I have only one minor comment left from my first review (which, retrospectively, was somewhat too critical – my apologies for that):

• I read through the imputation strategy for the top-censored income in the supplementary information. All in all its convincing. My only request is concerned with the resulting uncertainty of imputed values: Do you adjust for the fact that these values are imputed, for example by adding uncertainty by means of random draws from the corresponding error distribution? If not, I would at least critically mention this in the main text.

Finally, some minor issues that I found were:

• In the footnotes of Tables 3 and 4, you still state the rho value and the corresponding standard deviations of the multilevel models (for the first model, I guess), which are no longer needed since you include this information at the bottom in the main tables.

• You include year fixed effects in all the models but do not interpret them. What do they tell us? For the models of mean full-time earnings: Do they just reflect inflation? Or are earnings inflation adjusted? More substantially for the models regarding within-firm inequality: What do the significant and steadily increasing coefficients imply there? Maybe you could elaborate one or two sentences on that.

7. PLOS authors have the option to publish the peer review history of their article (what does this mean?). If published, this will include your full peer review and any attached files.

Reviewer #1: No

Reviewer #2: **Yes: **Christoph Zangger

---

## [Author Response · Author response to Decision Letter 1]

4 Aug 2020

Response to Reviewers

Reviewer 1 suggests a more clear statement of hypotheses and results in the abstract and one improvement in text.

These were great suggestions and we have changed the abstract as suggested and on page four changed “establishment level internal simplifications” to “reductions in occupational skill complexity”

Reviewer 2 asks “ Do you adjust for the fact that these values are imputed, for example by adding uncertainty by means of random draws from the corresponding error distribution? If not, I would at least critically mention this in the main text.”

We do not adjust for this uncertainty in our estimates, nor did previous research. It is an important technical point, and we will consider this as an improvement to future iterations of our imputation strategy. We now note in Supplemental Information text that this is a limitation:

1.4 Limitations

“This imputation is designed to produce a single point estimate or each top coded case. While this is unlikely to be a problem in very large samples, such as ours or Card et al (2013), future applications should consider recognizing uncertainty in the imputation with some form of multiple imputation strategy.”

Reviewer 2 suggests dropping Table 3 and 4 footnotes referring to Rho statistics. These footnotes refer to a null model not reported in the table not the reported estimates and so we prefer to keep them.

Finally, reviewer 2 asks: You include year fixed effects in all the models but do not interpret them. What do they tell us?

For Table 3 we add this interpretation: “We also note from the baseline model that mean workplace inflation adjusted earnings in Germany barely changed over time, rising slowly through 2001 and declining thereafter.”

For Table 4 : “The time trend reported in the baseline model confirms that average within workplace inequality has been rising across the study period.”

---

## [Editor Report · Decision Letter 2]

7 Aug 2020

The Organizational Production of Earnings Inequalities, Germany 1995-2010

PONE-D-20-03141R2

Dear Dr. Tomaskovic-Devey,

We’re pleased to inform you that your manuscript has been judged scientifically suitable for publication and will be formally accepted for publication once it meets all outstanding technical requirements.

Kind regards,

Semih Tumen, PhD

Academic Editor

PLOS ONE

---

## [Editor Report · Acceptance letter]

28 Aug 2020

PONE-D-20-03141R2 

The Organizational Production of Earnings Inequalities, Germany 1995-2010 

Dear Dr. Tomaskovic-Devey:

I'm pleased to inform you that your manuscript has been deemed suitable for publication in PLOS ONE. Congratulations! Your manuscript is now with our production department. 

Kind regards, 

on behalf of

Professor Semih Tumen 

Academic Editor

PLOS ONE